# First 72-hours after birth: Newborn feeding practices and neonatal mortality in India

**Piyasa Mal**[1], **Usha Ram**[2]*

1 Department of Public Health and Mortality Studies, International Institute for Population Sciences, Mumbai, Maharashtra, India, 2 Department of Biostatistics and Epidemiology, International Institute for Population Sciences, Mumbai, Maharashtra, India

* usharamagrawal@gmail.com

**Data Availability Statement:** Data is publicly available and accessible upon request from the DHS program at https://dhsprogram.com/data/available-datasets.cfm

**Funding:** The authors received no specific funding for this work.

## Abstract

### Background

The reductions in mortality levels among children under five years are observed in most populations, including populations that were lagging the progress in the past. However, the reduction is not uniform across ages during childhood. The mortality declines within the first month have shown relatively slow progress. Early initiation of breastfeeding and discarding pre-lacteal feed protects the newborn from acquiring infection and, thereby, reduces mortality. This paper assesses the change in the prevalence of early initiation of breastfeeding and pre-lacteal feed along with their associated factors, and their association with neonatal mortality in India.

### Methods

We used data from the three rounds of National Family Health Surveys conducted during 2005–06, 2015–16 and 2019–21 in India. We used bivariate and multivariate analyses to examine prevalence rates, risk factors, and relationships between breastfeeding practices, including early initiation of breastfeeding and pre-lacteal feed, and neonatal mortality.

### Results

Early initiation of breastfeeding within one hour after birth increased rapidly from 25% in 2005–06 to 42% in 2019–21, and the pre-lacteal feeding practice declined from 57% in 2005–06 to 15% in 2019–21. Pre-lacteal feed is lower in states/districts where early breastfeeding initiation is predominant and vice versa. The role of health professionals during pregnancy and the first two days after delivery significantly improved breastfeeding practice. Further, the findings suggest that an early breastfeeding initiation is associated with lower neonatal mortality, whereas pre-lacteal feed is not harmful compared to late breastfeeding initiation.

### Conclusion

Prevalence of pre-lacteal feed reduced, and initiation of early breastfeeding increased considerably after the launch of the National Rural Health Mission in India. However, after

**Competing interests:** The authors have declared that no competing interests exist.

2015–16, early breastfeeding initiation has stagnated, and the decline in pre-lacteal feed has slowed down. The future program needs special attention to emphasize the availability and accessibility of breastfeeding advisers and observers in health facilities to help mitigate adverse neonatal outcomes.

## Introduction

The past few decades have witnessed rapid improvements in the global health scenario as a result of childhood mortality reduction in most populations. Several countries, including India, for example, have made remarkable progress in mortality reductions among children below five years of age (under-five mortality rate, U5MR), especially during the past decade. The global child mortality rate has dropped by 60% between 1990 and 2020, from 93 to 37 per 1,000 live births [1]. However, the available evidence suggests that the reduction is slower in mortality within the first month of life. The United Nations Inter-agency Group for Child Mortality Estimation (UNIGME, 2021) estimates indicate that globally the neonatal mortality rate (NMR) has dropped from 37 per 1,000 live births in 1990 to 17 per 1,000 live births in 2020, yet contributed 47% of all under-five deaths [2].

In 2020, India accounted for the highest number of global neonatal deaths (490 thousand; 20% of global neonatal deaths) [3]. In India, neonatal deaths comprised almost half of all under-5 deaths in 2000, which rose 62% in 2020 [4]. Globally as well as in India, prematurity and low birth weight, neonatal infections (neonatal pneumonia, neonatal sepsis, and central nervous system infections), and birth asphyxia and birth trauma account for more than three-quarters of all neonatal deaths [5].

Early initiation of breastfeeding protects the newborn from acquiring infections and reduces mortality [6–8]. It further strengthens the emotional bonding between the newborn and mother and positively impacts the duration of exclusive breastfeeding [9, 10]. The yellow or golden first milk produced in the first few days (also called colostrum) is an essential source of nutrition and immune protection for the newborn. Colostrum is a complex biological nutrient-rich fluid produced by female mammals immediately after giving birth that provides immune, growth, and tissue repair capabilities [11, 12]. Early breastfeeding initiation can significantly contribute to the achievement of the child survival [13].

Breastfeeding among Indian mothers is nearly universal; however, myths and superstitions such as discarding colostrum, delayed initiation of breastfeeding, pre-lacteal feeding, and early initiation of complementary feeding persist [14, 15]. Several social, cultural, and economic factors, including maternal education and employment, maternal age, attitude and confidence, prenatal intention, ethnicity, residence, type of family, emulating western lifestyles, the influence of healthcare professionals, and availability of infant formula, influence breastfeeding practices [14, 16]. Further, breastfeeding attitudes and practices differ in different segments of society depending on traditional cultural practices and taboos prevalent in them [13]. Timely initiation of breastfeeding remains low in India, and the pre-lacteal feeding is still prevalent [17]. As also reflected from the global estimate (WHO 2021), 3 in 5 newborn babies in India are not breastfed within the first hour of life [18].

India is committed to achieving the targeted reduction in child mortality under the third Sustainable Development Goal (SDG) and is making multi-pronged efforts in achieving them. This study assesses breastfeeding practices regarding early initiation and pre-lacteal feeding in India and their association with neonatal mortality. The study specifically appraises i) the

prevalence of early initiation of breastfeeding and pre-lacteal feeding practices by socio-demographic characteristics of mother and child, and household; ii) predictors of early initiation of breastfeeding and pre-lacteal feeding practices; and iii) how the level of neonatal mortality differs across the status of pre-lacteal feeding and delayed breastfeeding initiation.

## Methods

### Data

We have used data from the last three rounds of the National Family Health Survey (NFHS-3 conducted during 2005–06, NFHS-4 conducted during 2015–16, and NFHS-5 conducted during 2019–21) to analyse trends in the early initiation of breastfeeding and pre-lacteal feed. The details of the survey are available at (https://dhsprogram.com/data/available-datasets.cfm). The NFHS-3 (2005–06) surveyed 109,041 households and 124,385 women of reproductive age. From NFHS-4 (2015–16), the survey was implemented at the district levels and interviewed 572,000 households and 699,686 women of reproductive age. The NFHS-5 (2019–21) surveyed 636,699 households and 724,115 women of reproductive age. Data were collected following all the ethical guidelines. Ethical approval was granted from the Institutional Review Board (IRB) of the International Institute for Population Sciences (IIPS) and ICF Institutional Review Board. During the field survey, verbal and written consent were taken from the respondents.

### Study sample

A total of 56,381 children born during the past five years preceding the survey were enumerated in 2005–06. Of these, 39,620 were the last-born surviving children and the information on pre-lacteal feed and timing of breastfeeding initiation was available for 38,527 children. In NFHS-4 (2015–16), 249,967 children born during the past five years preceding the survey were enumerated, and of them, 184,641 were the youngest surviving children. The information on pre-lacteal feed and timing of breastfeeding were available for 174,468 children. The NFHS-5 (2019–21) enumerated 230,870 children who were born during the past five years preceding the survey and out of these, 174,947 were the youngest surviving children. However, pre-lacteal feed and timing of breastfeeding initiation information were available for 165,131 children.

### Outcome variables

The study has two primary outcome variables: early initiation of breastfeeding and pre-lacteal feed. The NFHS asked women who had a livebirth during the past five years preceding the survey–'How long after birth did you start breastfeeding (your youngest surviving child)?' We considered the 'early initiation of breastfeeding' practice if mothers breastfed the child within one hour after birth. Similarly, the survey also asked mothers–'In the first three days after delivery, was the baby given anything to drink other than breast milk?' If mothers reported that they gave child anything other than breast milk, we considered it as 'pre-lacteal feed'. In addition, the neonatal mortality rate was estimated stratified by the status of breastfeeding initiation and pre-lacteal feeding. As mentioned previously, the study sample constitutes the last-born children of the women of reproductive age during the last five years preceding the respective surveys.

### Predictor variables

These predictor variables include: place of residence (rural, urban); sex of the child (male, female); maternal age at birth ($\leq$19, 20–34, $\geq$35); birth order (1, 2, 3 or more), maternal education (no education, primary, secondary or higher); household wealth tertile (poor, middle,

rich); maternal caste/tribe (Scheduled Castes (SCs), Scheduled Tribes (STs), Other Backward Class (OBC), others); mother's ANC visits during pregnancy (No visit, 1–3, 4 or more); place of delivery (not in a health facility, public health facility, private health facility); types of delivery (vaginal deliveries, C-section in a public facility, C-section in a private facility); mother received breastfeeding advice during pregnancy (no, yes), and healthcare provider observed breastfeeding during first two days after birth (no, yes).

## Statistical analysis

By applying appropriate sampling weights, we conducted bivariate analyses to assess the levels of early initiation of breastfeeding and pre-lacteal feeding practice by socioeconomic and demographic characteristics of the mother, child, and household. We used Chi-squared tests to assess statistical significance of bivariate associations of outcome variables with the socio-economic and demographic characteristics.

The study has two dichotomous outcome variables: early initiation of breastfeeding and pre-lacteal feed. The surveys collected information on these outcome variables only for the youngest surviving children. The information for older living children were not collected, and therefore, we applied logistic regression models to examine the odds of early initiation of breastfeeding and pre-lacteal feeding practice concerning exposure variables using the pooled datasets of the last three rounds of the NFHS covering the past two decades. We tested all exposure variables for possible multi-collinearity before inclusion into the regression models.

Two studies, viz. Edmond et al. [19] and Phukan et al. [20] have examined impact of breastfeeding practices on child mortality. While Edmond et al. [19] found that pre-lacteal feeding increased infant mortality in rural Ghana. Phukan et al. [20] observed that neonatal mortality was higher among neonates with delayed initiation of breastfeeding in India. Given the paucity of studies, the present study conducted a bivariate analysis to examine the association between levels of neonatal mortality by timing of initiation of breastfeeding and pre-lacteal feeding practices in Indian population using the data for the period 2019–21. Additionally, we used scatter plots to explore the ecological (at state level) association between the level of NMR and the breastfeeding and pre-lacteal feeding practices.

We conducted the analyses using STATA version 16 [21]. Maps were created using ArcMap 10 [22]. The shape files of the maps were accessed from the official website of the Demographic Health Survey (https://dhsprogram.com/data/available-datasets.cfm).

## Results

### Levels and trends in early initiation of breastfeeding and pre-lacteal feeding: National, sub-national and by population subgroups

Early initiation of breastfeeding within one hour after birth increased rapidly from just about one-quarter in 2005–06 to nearly 42% in 2019–21 and the percentage of newborn babies given a pre-lacteal feed declined from more than 57% in 2005–06 to 15% in 2019–21 (Fig 1).

In sixteen of the 30 Indian states in 2005–06, fewer than 35% of newborn babies were fed breastmilk immediately after birth. However, in 11 states (Maharashtra, Kerala, Tamil Nādu, Orissa, and the north-eastern states), more than 50% of the newborn babies were breastfed immediately after birth (data not shown). The early initiation of breastfeeding improved dramatically over time (Fig 2). By 2019–21, more than half of the newborn babies were breastfed within one hour of birth in 37% of districts, and in another 28% of districts 35–50% of newborn babies received breastfeeding within one hour of birth. Nonetheless, fewer than 35% of

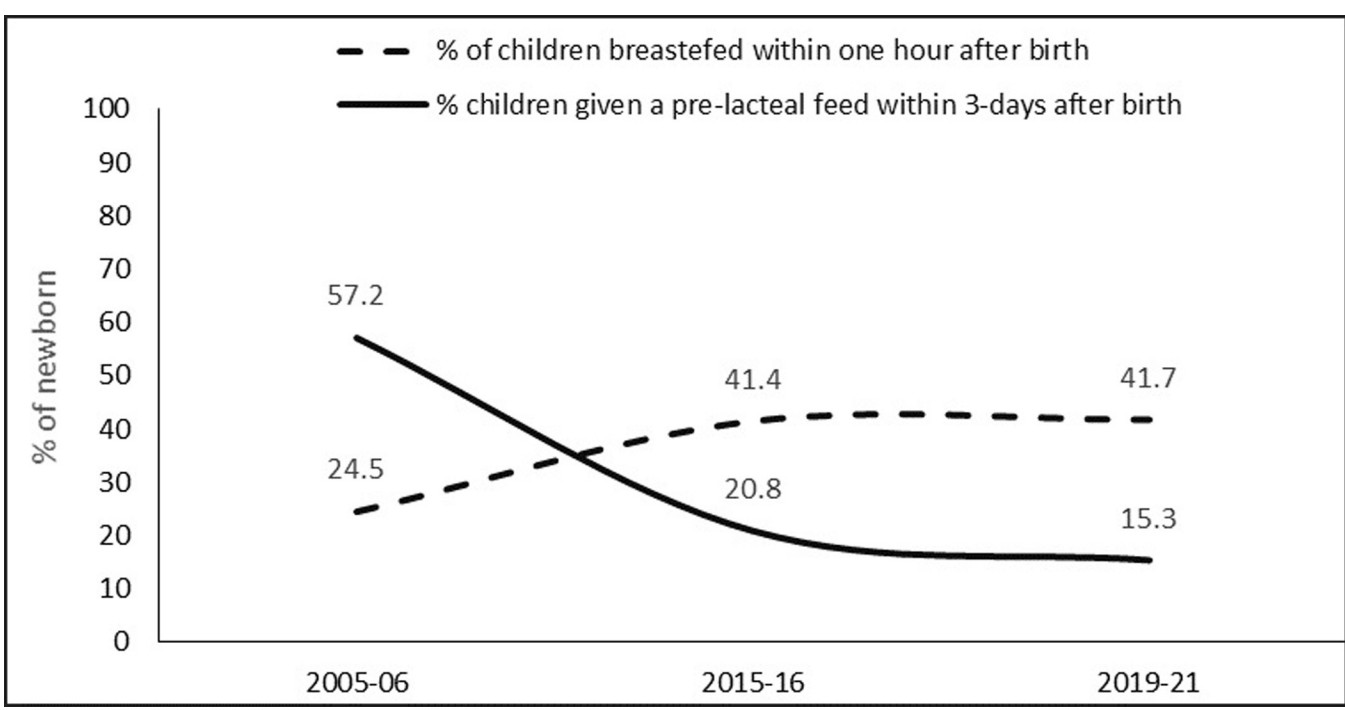

**Fig 1. Early initiation of breastfeeding and pre-lacteal feed in India, 2005–06 to 2019–21.**

newborn babies were breastfed early in about 35% of districts, mostly in Uttar Pradesh, Bihar, Jharkhand, Rajasthan, and Gujarat.

Pre-lacteal feed showed a downward trend in all the states/districts of India. In 2005–06, while more than 25% of newborn babies were given a pre-lacteal feed in 23 states, including Uttar Pradesh, Bihar, Punjab, Rajasthan, Madhya Pradesh, West Bengal etc, fewer than 12.5% of newborn babies were given a pre-lacteal feed in Kerala and Sikkim (data not shown). By 2019–21, the proportions reduced tremendously. For example, more than 25% of newborn babies received a pre-lacteal feed in about 11% of the Indian districts (mostly concentrated in Uttar Pradesh). In nearly 46% of the districts in 2019–21, fewer than 12.5% of newborn babies received a pre-lacteal feed.

Compared to the children born to mothers in rural areas, children in urban areas had an advantage, as higher proportion of children born to urban mothers were initiated early breast-feeding and fewer were given a pre-lacteal feed(Table 1). For example, in 2019–21, 44% of newborn babies in urban areas were breastfed within one hour compared to 41% newborn babies in rural areas. Similarly, 15% of newborn babies in urban areas were given a pre-lacteal feed compared 17% of newborn babies in rural areas. The feeding practice did not differ by newborn' gender. While early initiation of breastfeeding was higher and, pre-lacteal feed was lower among children born to younger mothers and children of lower birth order (1 or 2). Similarly, children born to mothers with secondary or higher education, belonging to rich wealth tertile had huge advantage over those born to mothers with no education or mothers in poor wealth tertile as more received early breastmilk and fewer were given a pre-lacteal feed. For example, in 2019–21, 37% of children born to mothers who did not receive any education and 43% children born to mothers who had completed secondary or higher education received early breastmilk. The corresponding numbers for children receiving a pre-lacteal feeding are 15% and 16%, respectively. A higher proportion of children born to scheduled tribe mothers

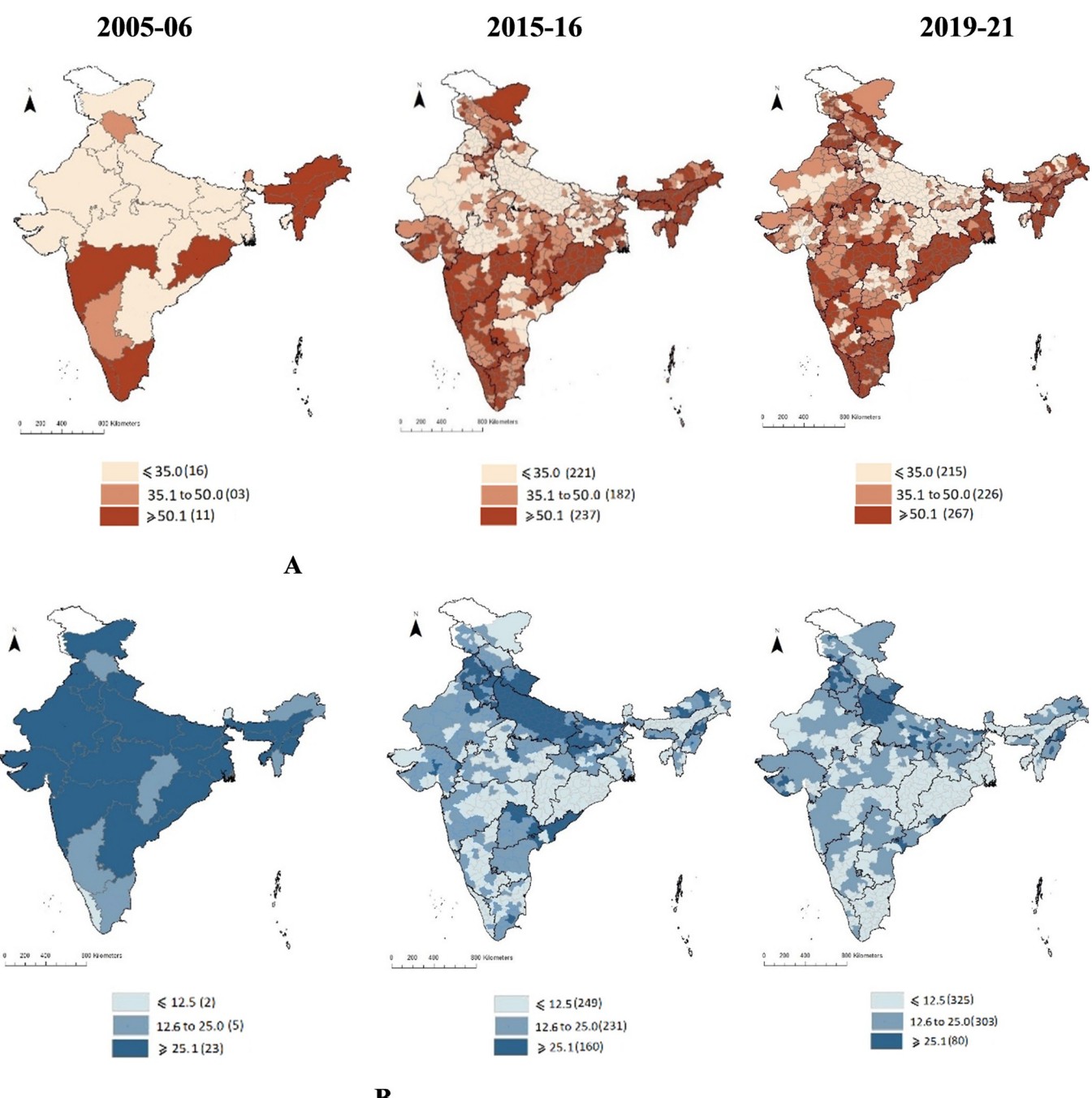

**Fig 2. Early initiation of breastfeeding (within one hour after birth) and pre-lacteal feed after birth across districts in India, 2015–16 to 2019–21.** A: Prevalence of early initiation of breastfeeding, B: Prevalence of pre-lacteal feed. Note: NFHS 3 (2005–06) provides statewise data only.

received breastmilk early while fewer received a pre-lacteal feed compared to the children born to mothers of other caste groups.

In 2019–21, early initiation of breastfeeding was higher among children whose mothers received 4 or more ANC visits (47%) than those with fewer ANC visits (35%) or no ANC visit (32%). Conversely, fewer children (15%) were given a pre-lacteal feed if the mothers received 4

**Table 1. Percentage of children* who were breastfed within one hour after birth and who received a pre-lacteal feeding by selected socio-demographic characteristics, India, 2005–06 to 2019–21.**

| | Early initiation of breastfeeding | | | Pre-lacteal feeding | | |
|---|---|---|---|---|---|---|
| | 2005–06 | 2015–16 | 2019–21 | 2005–06 | 2015–16 | 2019–21 |
| **Overall** | **24.5** | **41.4** | **41.7** | **57.2** | **20.8** | **15.3** |
| Place of residence | | | | | | |
| Rural | 22.4 | 40.9 | 40.7 | 57.6 | 20.6 | 17.1 |
| Urban | 30.3 | 42.6 | 44.2 | 48.0 | 21.2 | 14.6 |
| Sex of child | | | | | | |
| Male | 24.7 | 41.0 | 41.7 | 55.2 | 21.0 | 15.6 |
| Female | 24.3 | 41.9 | 41.7 | 54.9 | 20.5 | 14.9 |
| Maternal age at birth (in years) | | | | | | |
| ≤19 | 22.3 | 42.6 | 45.7 | 56.9 | 18.7 | 14.0 |
| 20–34 | 25.3 | 41.6 | 41.6 | 53.9 | 20.7 | 15.3 |
| ≥35+ | 18.3 | 36.6 | 38.3 | 66.5 | 25.6 | 16.0 |
| Birth order | | | | | | |
| 1 | 26.5 | 41.2 | 41.1 | 52.0 | 22.0 | 18.0 |
| 2 | 30.0 | 43.9 | 43.8 | 47.2 | 18.0 | 14.0 |
| 3 or higher | 19.8 | 39.0 | 39.9 | 61.9 | 22.5 | 13.7 |
| Maternal education | | | | | | |
| No education | 16.7 | 35.9 | 36.8 | 65.0 | 23.8 | 14.7 |
| Primary | 26.1 | 40.6 | 41.9 | 51.7 | 20.4 | 13.5 |
| Secondary or higher | 33.4 | 44.2 | 43.1 | 44.1 | 19.5 | 15.8 |
| Household wealth tertile | | | | | | |
| Poor | 18.5 | 38.9 | 39.3 | 63.4 | 21.4 | 12.4 |
| Middle | 23.9 | 42.9 | 42.1 | 55.9 | 18.9 | 14.6 |
| Rich | 30.4 | 42.3 | 43.4 | 46.8 | 22.0 | 18.4 |
| Maternal caste/tribe | | | | | | |
| Scheduled castes | 23.2 | 41.2 | 41.9 | 56.7 | 19.7 | 14.2 |
| Scheduled tribe | 28.6 | 45.4 | 46.1 | 42.0 | 12.4 | 10.2 |
| Other backward class | 22.0 | 40.1 | 38.8 | 60.4 | 23.4 | 16.1 |
| Other castes | 26.8 | 41.2 | 43.6 | 51.7 | 22.2 | 18.8 |
| Mother received ANC visits during pregnancy | | | | | | |
| No visit | 11.3 | 34.0 | 32.4 | 75.3 | 23.8 | 15.4 |
| 1 to 3 visits | 21.1 | 37.2 | 34.9 | 58.1 | 23.3 | 16.5 |
| 4 or more visits | 35.8 | 46.4 | 46.5 | 40.0 | 18.4 | 14.7 |
| Place of delivery | | | | | | |
| Not in a health facility | 17.8 | 34.3 | 34.2 | 64.9 | 29.0 | 20.9 |
| Public health facility | 37.3 | 45.8 | 44.6 | 34.8 | 14.4 | 10.5 |
| Private health facility | 31.1 | 38.1 | 38.1 | 46.6 | 27.2 | 24.4 |
| Type of delivery | | | | | | |
| Vaginal deliveries | 24.8 | 43.4 | 43.8 | 55.3 | 18.8 | 12.1 |
| C-section public facility | 25.9 | 37.7 | 39.5 | 43.3 | 19.4 | 17.2 |
| C-section private facility | 20.1 | 30.9 | 32.1 | 56.5 | 34.7 | 31.2 |
| Mother received breastfeeding advice during pregnancy | | | | | | |
| No | 19.0 | 34.4 | 37.8 | 61.2 | 28.4 | 19.4 |
| Yes | 36.1 | 47.0 | 44.4 | 40.4 | 16.6 | 13.8 |
| Healthcare providers observed breastfeeding during first 2 days after birth | | | | | | |
| No | NA | NA | 36.3 | NA | NA | 18.4 |

*(Continued)*

**Table 1.** (Continued)

| | Early initiation of breastfeeding | | | Pre-lacteal feeding | | |
|---|---|---|---|---|---|---|
| | 2005–06 | 2015–16 | 2019–21 | 2005–06 | 2015–16 | 2019–21 |
| Yes | NA | NA | 43.4 | NA | NA | 14.6 |
| N | 38528 | 184641 | 174947 | 38528 | 174,468 | 165,131 |

Notes:

The results of χ2 suggest that all variables are significant at less than 1% for both early initiations of breastfeeding and pre-lacteal feeding for all three time periods included in the analysis.

Sex of the child is significant at <10% for early initiations of breastfeeding in 2019–21 and is insignificant for the years 2005–06 for both early initiations of breastfeeding and pre-lacteal feeding

NA = Not available

*youngest surviving child among children born during the past five years preceding the survey

or more ANC visits during pregnancy compared to those who received fewer (17%) or no ANC visit (15%). Relatively higher proportions of newborn babies were initiated early breastfeeding and fewer were given a pre-lacteal feed if the delivery occurred in a public health facility. In contrast, only fewer children were breastfed early and more received a pre-lacteal feed if the delivery did not occur in a health facility. Similarly, higher proportions of children were breastfed early (44%) and fewer received a pre-lacteal feed (12%) if the delivery was vaginal.

The percentage of children who were breastfed early was higher among mothers who were advised about early initiation of breastfeeding during the pregnancy. At the same time, fewer children whose mothers received this advice were given a pre-lacteal feed. Similarly, feeding practices were better when a healthcare provider observed breastfeeding during the first 2 days after birth.

Not only have the feeding practices improved in all population subgroups during the analysis period, but also the gap across population subgroups too have narrowed. For example, the relative disadvantage in rural children has reduced from 8% points in 2005–06 to 3% points in 2019–21. Similarly, the relative disadvantage of pre-lacteal feed reduced from about 10% points to 3% points during the same period.

**Pre-lacteal feed by delivery type.** Regardless of the time period, other milk (not breastmilk) followed by plain water, honey and Sugar, glucose or salt water solution was the most common pre-lacteal feed (Table 2). About 59% of newborn babies in 2005–06 were fed with

**Table 2. Distribution of children by types of pre-lacteal feed and delivery type, India, 2005–06 to 2019–21.**

| Pre-lacteal type | 2005–06 | | | 2015–16 | | | 2019–21 | | |
|---|---|---|---|---|---|---|---|---|---|
| | Vaginal | C-section | Overall | Vaginal | C-section | Overall | Vaginal | C-section | Overall |
| Other milk (not breastmilk) | 59.1 | 54.4 | 58.7 | 61.7 | 69.6 | 63.8 | 68.6 | 77.1 | 71.9 |
| Plain water | 15.8 | 15.8 | 15.8 | 12.6 | 11.9 | 12.4 | 12.7 | 8.7 | 11.2 |
| Honey | 25.5 | 16.8 | 24.7 | 15.2 | 9.0 | 13.6 | 10.5 | 6.4 | 8.9 |
| Sugar/glucose/salt water or solution | 26.1 | 30.3 | 23.2 | 11.8 | 12.9 | 12.1 | 8.5 | 7.2 | 8.0 |
| *Janam ghutti* | 8.9 | 3.9 | 8.4 | 8.4 | 4.6 | 7.4 | 7.0 | 3.2 | 5.5 |
| Tea/infusions | 5.6 | 2.6 | 5.3 | 7.2 | 2.1 | 5.8 | 7.4 | 2.3 | 5.4 |
| Infant formula | 0.6 | 7.1 | 1.2 | 2.0 | 6.6 | 3.2 | 2.6 | 7.6 | 4.6 |
| Fruit juice | 0.2 | 0.2 | 0.2 | 0.8 | 0.7 | 0.8 | 1.9 | 1.2 | 1.6 |
| Gripe water | 0.5 | 1.1 | 0.6 | 1.4 | 1.9 | 1.5 | 1.5 | 1.6 | 1.5 |
| Others | 3.8 | 1.8 | 3.6 | 4.0 | 2.6 | 3.6 | 3.2 | 3.3 | 3.3 |

**Table 3. Adjusted odds ratios (and 95% confidence intervals) from logistic regression analyses examining children's likelihood of receiving early initiation of breastfeeding and pre-lacteal feed by selected characteristics, India [Based on Pooled NFHS Data, 2005–06, 2015–16, 2019–21].**

| Characteristics | Odds Ratio (95% confidence interval) | |
| --- | --- | --- |
| | Early initiation of breastfeeding | Pre-lacteal feed |
| Place of residence | | |
| Rural Ⓡ | | |
| Urban | 1.10***[1.06,1.13] | 0.93***[0.89,0.97] |
| Sex of child | | |
| Female Ⓡ | | |
| Male | 0.99[0.97,1.01] | 1.04*[1.01,1.07] |
| Maternal age at birth (in years) | | |
| ≤19 Ⓡ | | |
| 20–34 | 0.90***[0.86,0.95] | 1.06[1.00,1.13] |
| ≥35 | 0.88***[0.82,0.95] | 1.17**[1.07,1.29] |
| Birth order | | |
| 1 Ⓡ | | |
| 2 | 1.20***[1.16,1.23] | 0.73***[0.70,0.76] |
| 3 or higher | 1.11***[1.08,1.15] | 0.82***[0.79,0.86] |
| Maternal education | | |
| No education Ⓡ | | |
| Primary | 1.18***[1.13,1.23] | 0.86***[0.81,0.90] |
| Secondary or higher | 1.30***[1.26,1.35] | 0.76***[0.72,0.79] |
| Household wealth tertile | | |
| Poor Ⓡ | | |
| Middle | 1.01[0.97,1.04] | 1.10***[1.05,1.15] |
| Rich | 1.00[0.96,1.04] | 1.24***[1.18,1.31] |
| Maternal caste/tribe | | |
| Scheduled Tribe Ⓡ | | |
| Scheduled Castes | 0.84***[0.81,0.88] | 1.64***[1.54,1.75] |
| Other backward class | 0.78***[0.75,0.81] | 1.80***[1.70,1.91] |
| Other castes | 0.84***[0.81,0.88] | 1.84***[1.72,1.97] |
| Mother received ANC visits during pregnancy | | |
| No visit Ⓡ | | |
| 1 to 3 visits | 0.98[0.93,1.03] | 1.18***[1.11,1.26] |
| 4 or more visits | 1.53***[1.45,1.62] | 0.84***[0.78,0.89] |
| Place of delivery | | |
| Not in a health facility Ⓡ | . | |
| Public health facility | 1.45***[1.40,1.51] | 0.46***[0.44,0.48] |
| Private health facility | 1.19***[1.14,1.25] | 0.82***[0.77,0.87] |
| Type of delivery | | |
| Vaginal deliveries Ⓡ | | |
| C- section in a public facility | 0.64***[0.61,0.67] | 1.81***[1.71,1.92] |
| C- section in a private facility | 0.56***[0.53,0.59] | 2.12***[2.00,2.25] |
| Mother received breastfeeding advice during pregnancy | | |
| No Ⓡ | | |
| Yes | 1.44***[1.40,1.49] | 0.56***[0.54,0.59] |
| Survey period | | |
| 2005–06 Ⓡ | | |
| 2015–16 | 1.61***[1.52,1.70] | 0.34***[0.32,0.36] |

*(Continued)*

**Table 3.** (Continued)

| Characteristics | Odds Ratio (95% confidence interval) | |
|---|---|---|
| | Early initiation of breastfeeding | Pre-lacteal feed |
| 2019–21 | 1.43***[1.35,1.51] | 0.27***[0.25,0.29] |

Ⓡ = Reference category
* p<0.05
** p<0.01
*** p<0.001

other milk (not breastmilk), which increased to 72% in 2019–21. About 11% and 9% of newborn babies in 2019–21 were fed plain water and honey, respectively. Another nearly 8% of newborn babies in 2019–21 were given sugar, glucose or saltwater solution before initiating breastfeeding. A minority of newborn babies were also given *janam ghutti*, tea/infusions, infant formula or gripe water. The choice of pre-lacteal feed varied by delivery type. For example, in 2019–21 higher percentages of newborn babies born by a C-section delivery (77%) were given other milk than vaginal delivery (69%). Likewise, a higher proportion of newborn babies delivered by a C-section reportedly were given plain water or honey before breastmilk.

## Determinants of early initiation of breastfeeding

After adjusting for other socioeconomic, demographic, and programmatic variables, results from logistic regression analysis (Table 3) reveal that compared to children born to rural mothers, children born to a mother in urban areas had higher odds of receiving breastmilk within one hour of birth (odds ratio (OR): 1.10 (95% CI: 1.06,1.13)). The odd of receiving early breastmilk were lower for children born to mothers aged 20–34 years and older than those whose mothers were older than 35 years had lower odds (OR: 0.90 (0.86,0.95) and 0.88 (0.82,0.95), respectively) than those born to younger mothers. Compared to first born, children of subsequent birth orders had higher odds to be breastfed within one hour after birth. Mother's educational attainment had a statistically significant positive association with early initiation of breastfeeding. For example, compared to newborn babies of uneducated mothers, newborn babies whose mothers had primary or secondary or higher education had higher odds (OR: 1.18 (1.13,1.23) and 1.30 (1.26,1.35), respectively) to have received early breastmilk. Compared to the scheduled tribes, the odds of receiving early breastmilk were lower for newborn born to mothers in other backward class (OR: 0.78 (0.75,0.81)) or to those in scheduled castes or other castes categories (OR: 0.84 (0.81,0.88)).

Newborn babies of mothers who had 4 or more ANC visits had substantially higher odds (OR: 1.53 (1.45,1.62)) of receiving early breastmilk than those born to mothers who had no ANC visit. Similarly, newborn babies delivered in a public health facility had higher odds (OR: 1.45 (1.40,1.51)) of receiving early breastmilk than those delivered in a non-health facility. Similarly, the odds were also higher for children born in a private health facility (OR: 1.19 (1.14,1.25)). Children delivered by a C-section in either a public or a private health facility had lower odds (OR: 0.64 (0.61,0.67) and 0.56 (0.53,0.59), respectively) of early initiation of breastfeeding than a vaginal delivery. Children born to mothers who received breastfeeding advice during pregnancy had much higher odds (OR: 1.44 (1.40,1.49)) of early breastfeeding than those born to mothers who were not advised. Similarly, odds of early initiation of breastfeeding were higher among children surveyed in 2015–16 (OR: 1.61 (1.52,1.70)) and 2019–21 (OR: 1.43 (1.35,1.51)) compared to those surveyed earlier (in 2005–06).

## Determinants of pre-lacteal feed

Results adjusted for other socioeconomic, demographic, and programmatic variables suggest that the odds of newborn babies getting a pre-lacteal feed were lower for those born to mothers in urban areas (OR: 0.93 (95% CI: 0.89,0.97)) than those born in rural areas. Newborn boys had higher odds to receive a pre-lacteal feed than the newborn girls (OR: 1.04 (1.01,1.07)). Newborn babies of older mothers had substantially higher odds (OR: 1.17 (1.07,1.29)) to get a pre-lacteal feed than those born to younger mothers. Compared to first born, children of subsequent birth order had lower odds of getting a pre-lacteal feed (OR: 0.73 (0.70,0.76) for second order births and 0.82 (0.79,0.86) for three or higher order births). Compared to newborn babies of uneducated mothers, newborn babies whose mothers had primary (OR: 0.76 (0.72,0.79)) or secondary or higher education (OR: 0.86 (0.81,0.90)) had much lower odds of receiving a pre-lacteal feed. Newborn babies of mothers belonging to middle and rich wealth tertile had significantly higher odds (OR: 1.10 (1.05,1.15) for middle wealth tertile and 1.24 (1.18,1.31) for rich wealth tertile) of receiving a pre-lacteal feed compared to those born to mothers in poor wealth tertile. Compared to the Scheduled tribes, newborn babies from other castes had substantially higher odds (OR: 1.64 (1.54,1.75) to 1.84 (1.70,1.91)) of receiving a pre-lacteal feed.

While newborn babies of mothers who had 4 or more ANC visits had lower odds (OR: 0.84 (95% CI: 0.78,0.89)) of receiving a pre-lacteal feed, those born to mothers who had 1–3 ANC visits had higher odds (OR: 1.18 (1.11,1.26)) of receiving a pre-lacteal feed than those born to mothers who had no ANC visit. Further, newborn delivered in a public health facility or in a private health facility had significantly lower odds (OR: 0.46 (0.44,0.48) and 0.82 (0.77,0.87), respectively) of receiving a pre-lacteal feed than those delivered outside a health facility. Children born to mothers who had a C-section delivery in a public or private health facility had much higher odds (OR: 1.81 (1.71,1.92) and 2.12 (2.00,2.25), respectively) of receiving a pre-lacteal feed than vaginal delivery. Newborn babies whose mothers received breastfeeding advice during pregnancy had much lower odds (OR: 0.56 (0.54,0.59)) of receiving a pre-lacteal feed than those whose mothers were not advised. Compared to children surveyed in 2005–06, odds of receiving a pre-lacteal feed was lower among children surveyed in 2015–16 (OR: 0.34 (0.32,0.36)) or in 2019–21 (OR: 0.27 (0.25,0.29)).

**Table 4. Neonatal mortality rate per thousand live births by the breastfeeding and pre-lacteal feeding status, India (2019–21).**

| Practices | Neonatal mortality rate (NMR) | Number of live births |
|---|---|---|
| Ever breastfed | | |
| Yes | 6.9 [6.5,7.3] | 1,67,606 |
| No | 197.7 [191.3, 206.8] | 16,377 |
| Did not answer | 28.4 [26.9, 29.9] | 46,887 |
| Timing of breastfeeding initiation | | |
| Within one hour | 6.6 [6.0, 7.2] | 72,960 |
| One hour to one day | 6.8 [6.3,7.4] | 77,001 |
| More than one day | 8.1 [6.7,9.6] | 15171 |
| Pre-lacteal feeding (for last birth) | | |
| Yes | 5.1 [4.3,6.1] | 25,212 |
| No | 7.1 [6.7,7.6] | 1,39,919 |
| Did not answer | 178.9 [171.4, 186.6] | 9,816 |
| Overall | 24.8 [24.17, 25.44] | 2,30,870 |

## Neonatal mortality and feeding practices

The overall estimated neonatal mortality rate (NMR) was 24.8 per 1000 livebirths in India in 2019–21 (Table 4). The NMR among children who were ever breastfed was 6.9 per 1000 live-births as against of 198 per 1000 livebirths among those who were never breastfed. The NMR was 28.4 per 1000 livebirths among children born to mothers who did not answer on breastfeeding status. The NMR was 6.6 and 6.8 per thousand livebirths among children who were breastfed within one hour after birth or between one hour to one day after birth, respectively. In contrast, the NMR was 8.1 per 1000 livebirths among children who received delayed breastfeeding one day after birth. The NMR was 5.1 per thousand livebirths among newborn who received pre-lacteal feed and 7.1 per 1000 live births among those who did not receive a pre-lacteal feed.

We also explored the ecological (at state level) association between the level of NMR and the breastfeeding practices, which is illustrated in Fig 3. The correlation between NMR and early initiation of breastmilk was -0.59 and that with pre-lacteal feed was 0.12 (data not shown). The results suggest that the NMR was lower in states where higher proportions of newborn babies received early breastmilk. However, the association between NMR and pre-lacteal feed appears relatively weak. These ecological associations are consistent with the pattern shown in Table 4.

## Discussion

The study documents a spatial pattern in initiation of early breastfeeding and pre-lacteal feed in India. The pre-lacteal feeding practice is less common in regions where early initiation of breastfeeding is predominant and vice versa. The study observed that early breastfeeding initiation is more commonly prevalent in southern India where pre-lacteal is less prevalent, conversely, initiation of early breastfeeding is lower and pre-lacteal is higher in the northern India. Mothers who gave a pre-lacteal feed, initiated breastfeeding later than the recommended guidelines [17, 23, 24]. The early breastfeeding initiation may actually help reduce pre-lacteal feeding. The breastfeeding practices in India have shown considerable improvement, especially after the launch of the National Rural Health Mission in 2005. However, the recent data shows a stagnation in the levels of early breastfeeding initiation. At the same time, the pace of decline in the pre-lacteal feeding too has slowed down.

The study found that the children delivered in the public health facilities have a higher chance of getting breastfeeding within one hour after birth. In contrast, the practice is less common for children born in private health facilities and is lowest for children not delivered in a health facility. Earlier studies have also found significant association between place of birth with timely initiation of breastfeeding [25, 26]. Further, pre-lacteal feeding is less common among children born in public health facilities and is relatively at higher levels for children born in private health facilities. This suggests that the service providers in public health facilities follow the breastfeeding guidelines more strictly than those in private health facilities. The levels of timely initiation of breastfeeding were found to be undesirably low, and the practice of giving pre-lacteal feeding existed even in tertiary-care hospitals [17].

The factors associated with timely initiation were higher maternal education, antenatal counselling, absence of obstetric problems, and vaginal delivery. In contrast, those associated with pre-lacteal feeding were lower maternal education and caesarean delivery [17, 27–29]. A study found that breastfeeding initiation was delayed after birth because the mothers perceived colostrum to be harmful for child's health and that the mother's milk is 'not ready' until two to three days postpartum [30].

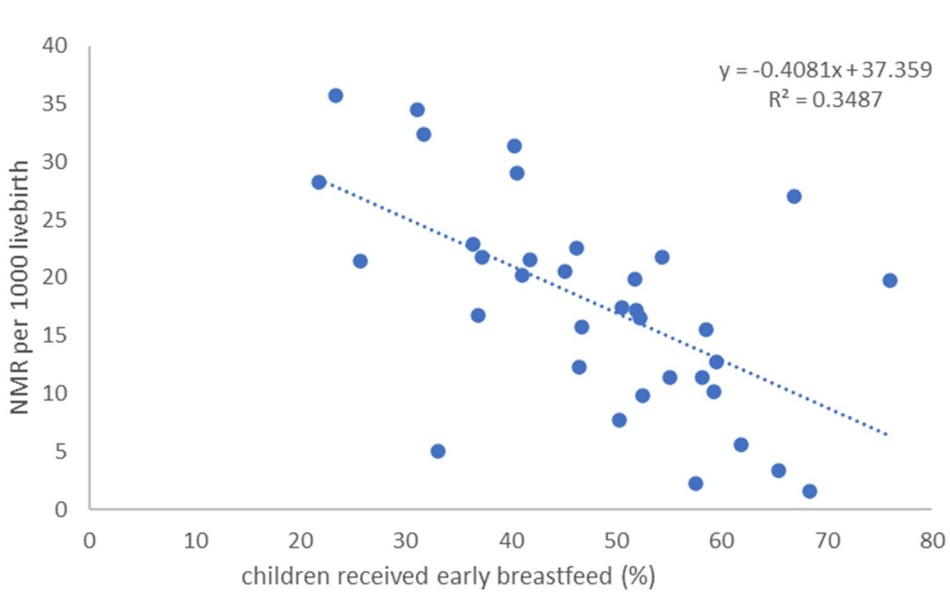

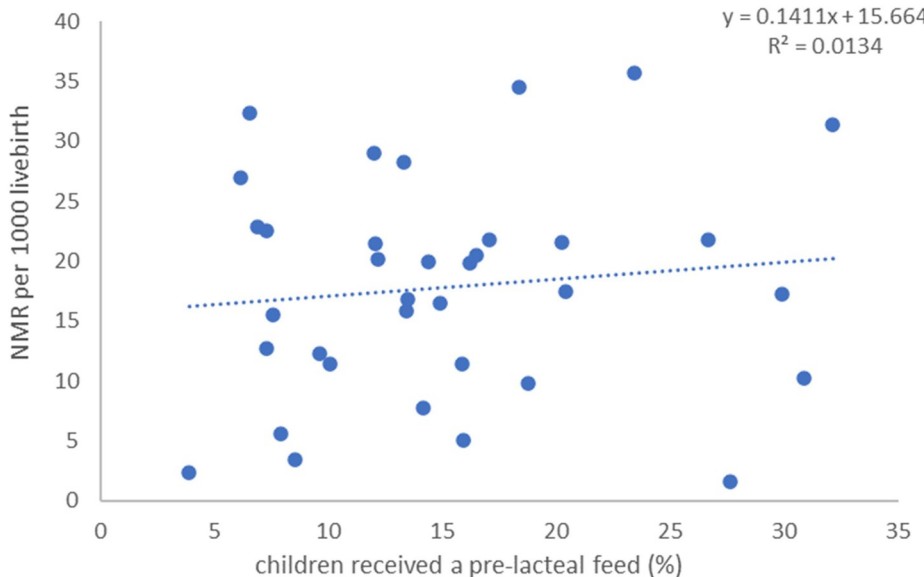

**Fig 3.** Association of neonatal mortality with early initiation of breastfeeding (A) and pre-lacteal feeding (B), based on state-level estimates, India, 2019–21.

The prevalence of pre-lacteal is significantly high, and early initiation is low in C-section deliveries. Use of general or spinal anaesthesia for caesarean delivery and the trauma during surgery delay mothers' recovery, which delays the early initiation of breastfeeding. Meanwhile, it is compensated by milk other than breastmilk [17, 31, 32]. Results show that the distribution of those items for which mothers' breastmilk in critical conditions can compensate is more predominant, such as milk other than breastmilk and infant formula in C-section delivery. But in vaginal deliveries, other factors related to cultures such as honey and *janam ghutti* were more dominant. The study also notes a change in the distribution of the types of pre-lacteal

feed over the study period. Earlier there was a predominance of items related to cultural and traditional practices. With time, the distribution has shifted in favour of honey, *janam ghutti*, sugar/glucose water is reduced, and milk other than breastmilk and infant formula.

The role of health professionals during pregnancy and the first two days after delivery is visible in improving breastfeeding practices. The study found that mothers who received breastfeeding advice during pregnancy have elevated levels of early initiation of breastfeeding and lower levels of pre-lacteal feeding practice. The study revealed that higher percentages of children were breasted within one hour of birth and lower percentages were given a pre-lacteal feed when healthcare providers observed breastfeeding during the first two days after birth. There is a significant association between two important interventions of receiving prenatal counselling on breastfeeding and postnatal support to mothers, with early initiation of breastfeeding [25, 33].

The study results show that early initiation of breastfeeding lowers neonatal mortality and thus provides compelling evidence supporting the critical role the timing of breastfeeding initiation, particularly within the first one hour can significantly reduce the risk of severe illnesses during the first month of life. Earlier studies also showed an increased risk of neonatal death by 33% when the breastfeeding initiation is delayed between 2 and 23 hours and doubles when the initiation is delayed beyond 24 hours [8, 34, 35]. This could be linked to the immune system which is still growing during infancy, making them highly vulnerable to diseases [36]. Breastmilk contains vital nutrients, antibodies, and other bioactive substances that work as a natural defence against illnesses and boost the immunological response during infancy [37]. Further, using the most recent available data of 2019–21, this study found that pre-lacteal feed does not influence child survival much. This could be due to the fact that the study found that the items given as pre-lacteal feed predominantly used in India are typically not harmful and can be compensated by mothers' breastmilk.

We also acknowledge a few limitations of this study. First, this study considered any food given to the newborn within the first three days after birth other than breastmilk as pre-lacteal feed. This is mainly due to the fact that the survey collects information by asking 'In the first three days after delivery, was the baby given anything to drink other than breast milk?' Further, the survey does not provide information on timing. Second, the pre-lacteal and initiation of breastfeeding information is available only for the last birth in the NFHS data. Due to this, the estimated NMR is restricted only to the last birth. Third, the NFHS 3 provides only state level data and hence unable to provide district level analysis.

## Conclusions

The present study found that promoting early initiation of breastfeeding reduces the pre-lacteal feeding practices suggesting that a stringent focus on enhancing levels of early initiation of breastfeeding may yield reductions in levels of pre-lacteal feeding among the newborn babies. The study findings further indicate that the breastfeeding advice during pregnancy and breastfeeding observation by the health service provider after the birth significantly improve breastfeeding and pre-lacteal feeding practices. This supports that increased focus on these practices by the grass root level heath workers at the time of visits can help accelerate reductions in neonatal mortality. However, there is a gap in the program implementation as it lacks universal adherence, more specifically in the private health facilities. Thus, it is of utmost importance that the future programs make concerted efforts and pay greater attention to ensure that the expectant mothers are duly advised on early initiation of breastfeeding and that the service providers observe the early initiation to ensure adherence. For example, it may be helpful to create the awareness and greater commitments among the service provides about the

importance of early initiation and its adherence and how this can bring about positive changes in the health outcomes among newborn babies including reduced mortality. Additionally, supporting new mothers with proper guidance and resources may facilitate establishing successful breastfeeding routines.

The study also documents evidence on association between neonatal mortality and breastfeeding and pre-lacteal feeding practices in India. The neonatal mortality rate is found to be lower among newborn who were given breastmilk within one hour after birth. The study further noted, that the pre-lacteal feed did not influence neonatal survival, but early initiation of breastfeeding does. Although, pre-lacteal feeding practices may continue given cultural context of the practices, it may be essential to create awareness on the choice of pre-lacteal feeds that are harmless. It is suggested that there is a need to strengthen the intervention of early breastfeeding initiation through breastfeeding adviser and cohesive observance of the early initiation to reduce neonatal mortality.

## Acknowledgments

We acknowledge and are grateful to anonymous reviewers and the academic editor for their elaborate, critical and constructive suggestions in improving the paper.

## Author Contributions

**Conceptualization:** Usha Ram.

**Formal analysis:** Piyasa Mal.

**Methodology:** Piyasa Mal.

**Supervision:** Usha Ram.

**Writing – original draft:** Piyasa Mal.

**Writing – review & editing:** Usha Ram.

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
