## [Decision Letter · Decision Letter 0]

24 Jul 2023

PONE-D-22-34236First 72-hours after birth: Newborn feeding practices and neonatal mortality in IndiaPLOS ONE

Dear Dr. Ram,

Thank you for submitting your manuscript to PLOS ONE. After careful consideration, we feel that it has merit but does not fully meet PLOS ONE’s publication criteria as it currently stands. Therefore, we invite you to submit a revised version of the manuscript that addresses the points raised during the review process.

We look forward to receiving your revised manuscript.

Kind regards,

Chandan Kumar, Ph.D.

Academic Editor

PLOS ONE

Journal Requirements:

"The authors did not receive any grant from any public, the commercial or non-profit funding agency for conducting this study."

5. We note that Figure 2 in your submission contain map images which may be copyrighted. All PLOS content is published under the Creative Commons Attribution License (CC BY 4.0), which means that the manuscript, images, and Supporting Information files will be freely available online, and any third party is permitted to access, download, copy, distribute, and use these materials in any way, even commercially, with proper attribution. For these reasons, we cannot publish previously copyrighted maps or satellite images created using proprietary data, such as Google software (Google Maps, Street View, and Earth). For more information, see our copyright guidelines: http://journals.plos.org/plosone/s/licenses-and-copyright.

(1) You may seek permission from the original copyright holder of Figure 2 to publish the content specifically under the CC BY 4.0 license.  

Reviewers' comments:

Reviewer's Responses to Questions

**Comments to the Author**

1. Is the manuscript technically sound, and do the data support the conclusions?

Reviewer #1: Yes

Reviewer #2: Yes

2. Has the statistical analysis been performed appropriately and rigorously? 

Reviewer #1: Yes

Reviewer #2: Yes

3. Have the authors made all data underlying the findings in their manuscript fully available?

Reviewer #1: Yes

Reviewer #2: Yes

4. Is the manuscript presented in an intelligible fashion and written in standard English?

Reviewer #1: Yes

Reviewer #2: Yes

5. Review Comments to the Author

Reviewer #1: The present paper is a timely contribution to the existing literature on neonatal mortality and determinants like initiation of breast feeding and the substance fed. However, while the paper highlights an important finding that ‘the pre-lacteal feed did not influence neonatal survival’; it may be important to consider (a) a couple of lines explanting the phenomenon; and (b) giving viable suggestions for strengthening early breastfeeding. This will add value to the paper. Similarly for lines 339-340 too, concrete suggestions will be useful for policy makers.

Reviewer #2: REPORT TEMPLATE

Paper title: First 72-hours after birth: Newborn feeding practices and neonatal mortality in India

Manuscript no: PONE-D-22-34236

Reviewer's report

The manuscript addressed trends, spatial variation, and determinants of early initiation of breastfeeding and pre-lacteal feeding in India. On very short, the manuscript also analyzed the effect of these breastfeeding practices on neonatal mortality. The study used the three latest rounds of the National Family Health Survey (NFHS) data conducted during 2005-06, 2015-16, and 2019-21. The study applied bivariate and multivariate (the Tobit regression analysis) analysis to examine the level, trends, and determinants of breastfeeding practices.

While reviewing this manuscript, I found that there is a mismatch between the title and the analysis presented in the manuscript. While the title indicates that this manuscript focused on what is the effect of breastfeeding practices on neonatal mortality. However, most of the result section focused on the level, trends, and determinants of the early initiation of breastfeeding and pre-lacteal feeding, and very less is talked about the effect of these on neonatal mortality. This is completely different than what the method section of the abstract claims. Therefore, my suggestion is please remove the neonatal mortality part and only focus on the level, trends, and determinants of the feeding practices. That itself will be a good enough material for a paper. Below is my specific observation on the paper.

Abstract:

1. In the method section, please mention the timing of the survey period. Some readers may wonder which three rounds, the first three or the last three out of the five rounds of the National Family Health Survey (NFHS). Though I noticed that it was mentioned in the results section, however, it’s always good to mention the survey date in the method section.

2. Please mention and define the outcome variable in the method section.

3. Please provide some numeric values in the result section.

4. Some parts of the result section can be moved into the discussion or in the conclusion. Such as “While the prevalence of pre-lacteal feed reduced and initiation of early breastfeeding increased considerably after the launch of National Rural Health Mission in India….”.

5. It would be good to present the results and conclusion separately.

Introduction:

6. Is the first sentence complete?? “The past few decades have witnessed rapid improvements in the global health scenario as a result of mortality reduction in most populations, including lagers”. Please check. Further, the very first sentence of the introduction section is talking about the reduction in overall mortality. Isn’t it?? If so, then my suggestion is please start talking about childhood mortality or neonatal mortality.

7. Please abbreviate ‘UNIGME’ as many readers may not be aware of it.

8. The “)” at the last of the first paragraph seems unnecessary, please check and remove it.

Data and method:

9. In the very first sentence, please write ‘2005-6’ as “2005-06”. Also please provide the survey period of each of the last three rounds of the NFHS.

10. Nothing was mentioned about neonatal mortality. Is neonatal mortality, not your dependent variable? You may consider ‘Early initiation’ as well as ‘Pre-lacteal feed’ as predictors, and independent variables as confounders or independent. Moreover, please provide some reference that why you are considering these independent variables in the analysis.

Results:

11. From line number 160-185, nowhere the results (numeric value) are reported. Please report the results in those paragraphs and shorten those.

12. In Table 3, for the reference categories, you have mentioned ‘@’ as well as ‘R’. Please keep only one.

Discussion:

13. Too much focus is on the early initiation of breastfeeding and pre-lacteal feeding, and almost nothing is talked about neonatal mortality.

Conclusions:

14. The conclusion section is focusing on neonatal mortality only.

6. PLOS authors have the option to publish the peer review history of their article (what does this mean?). If published, this will include your full peer review and any attached files.

Reviewer #1: **Yes: **Sanghmitra Acharya

Reviewer #2: No

---

## [Author Response · Author response to Decision Letter 0]

21 Aug 2023

Reviewer 1

Review of the Paper Titled: First 72-hours after birth: Newborn feeding practices and neonatal mortality in India

The present paper is of relevance in context of the child health in early years of life. While much of the literature concentrate on analysing the infant and child by the socio-cultural background of the mother. Initiation of breastfeeding as well as the substance to be fed is highly determined by it. The present paper mortality, the present paper engages with the crucial first 72 hours of life. This also the period which is influenced has used robust methodology to examine the feeding practices and neonatal mortality in India evident from the NFHS data of three selected rounds. The analysis bring out the nuances of the early neonatal stage and related mortality.

Response 1: We are very grateful to the reviewers for reviewing the manuscript and appreciating the work. We thank reviewer for constructive suggestions and comments. We have incorporated all these valuable suggestions in revised manuscript. 

The following points may be considered for value addition-

Comment 1:

In lines 28-29 as given below

28 The mortality within the first month of life (during the neonatal period) has shown relatively

29 slow progress

Observation/Suggestion� ‘Progress in mortality’ is certainly not what the authors intend. Therefore, this line needs to be re-stated as- The mortality decline within the first month…slow progress.

Response 2: Thanks for pointing this out. We have replaced ‘Progress in mortality’ with ‘mortality decline’. [please see line no. 27].

Comment 2:

In line 34 as given below

34 We used data from three rounds of National Family Health Surveys in India.

Observation/Suggestion� May be a good idea to mention the three rounds of NFHS used in the paper.

Response 3: Thanks for the suggestion. We have mentioned the last three rounds of NFHS. [please see line nos. 33-34]

Comment 3:

In line 332-334 as given below

332……… The study further noted, that the pre-lacteal feed did

333 not influence neonatal survival. It is suggested that there is a need to strengthen the intervention of early

334 breastfeeding initiation.

Observation/Suggestion� It is an important finding that ‘the pre-lacteal feed did not influence neonatal survival’. Therefore, (a) a couple of lines explanting the phenomenon; and (b) giving viable suggestions for strengthening early breastfeeding will add value to the paper. Similarly for lines 339-340 too, concrete suggestions will be useful for policy makers.

Response 4: Thanks for the suggestion. We have added a line in the revised manuscript and have explained the phenomenon [please see line nos. 398-400] and provided viable suggestions for strengthening early breastfeeding component for the future programme [please see line nos. 383-385, 400-402].

We have also provided suggestions that may be useful for policy makers [please see line nos. 389-393].

SANGHMITRA S ACHARYA -ORCID ID

https://orcid.org/0000-0001-6488-4181

Reviewer 2

Paper title: First 72-hours after birth: Newborn feeding practices and neonatal mortality in India

Manuscript no: PONE-D-22-34236

Reviewer's report

The manuscript addressed trends, spatial variation, and determinants of early initiation of breastfeeding and pre-lacteal feeding in India. On very short, the manuscript also analyzed the effect of these breastfeeding practices on neonatal mortality. The study used the three latest rounds of the National Family Health Survey (NFHS) data conducted during 2005-06, 2015-16, and 2019-21. The study applied bivariate and multivariate (the Tobit regression analysis) analysis to examine the level, trends, and determinants of breastfeeding practices.

While reviewing this manuscript, I found that there is a mismatch between the title and the analysis presented in the manuscript. While the title indicates that this manuscript focused on what is the effect of breastfeeding practices on neonatal mortality. However, most of the result section focused on the level, trends, and determinants of the early initiation of breastfeeding and pre-lacteal feeding, and very less is talked about the effect of these on neonatal mortality. This is completely different than what the method section of the abstract claims. Therefore, my suggestion is please remove the neonatal mortality part and only focus on the level, trends, and determinants of the feeding practices. That itself will be a good enough material for a paper. Below is my specific observation on the paper. 

Response 5: Thank you for your suggestion and comment. In the revised manuscript we have made necessary changes/modifications. However, the referred table and text, demonstrate that there is a strong association between neonatal mortality and breastfeeding practices as well as pre-lacteal feeding practice. Therefore, we have not removed neonatal mortality part.

Abstract: 

Comment 4:

In the method section, please mention the timing of the survey period. Some readers may wonder which three rounds, the first three or the last three out of the five rounds of the National Family Health Survey (NFHS). Though I noticed that it was mentioned in the results section, however, it’s always good to mention the survey date in the method section. 

Response 6: Thank you for your suggestion. We have mentioned the survey date in the method section of the abstract. [please see line nos. 33-34]

Comment 5:

Please mention and define the outcome variable in the method section. 

Response 7: Thank you for your suggestion. We have mentioned the outcome variable in the method section of abstract [please see line nos. 35-36]. Detailed definitions of the outcome variables are already mentioned in the method section of the complete manuscript [please see line nos. 106-112].

Comment 6:

Please provide some numeric values in the result section. 

Response 8: Thank you for your suggestion. We have included numeric values as necessary in the result section of the abstract. [please see line nos. 38-39]

Comment 7:

Some parts of the result section can be moved into the discussion or in the conclusion. Such as “While the prevalence of pre-lacteal feed reduced and initiation of early breastfeeding increased considerably after the launch of National Rural Health Mission in India….”.

Response 9: Thank you for your suggestion. We have moved this section into the conclusion. [please see line nos. 46-48]

Comment 8:

It would be good to present the results and conclusion separately.

Response 10: Thank you for your suggestion. We have moved this part under a new head ‘conclusion’. [please see line no. 45]

Introduction:

Comment 9:

Is the first sentence complete?? “The past few decades have witnessed rapid improvements in the global health scenario as a result of mortality reduction in most populations, including lagers”. Please check. Further, the very first sentence of the introduction section is talking about the reduction in overall mortality. Isn’t it?? If so, then my suggestion is please start talking about childhood mortality or neonatal mortality. 

Response 11: Thanks for pointing out. We have modified the sentence [please see line nos. 53-55]. We have replaced ‘mortality’ by ‘childhood mortality’ as suggested. [please see line no. 54]

Comment 10:

Please abbreviate ‘UNIGME’ as many readers may not be aware of it. 

Response 12: Thank you for your suggestion. We have mentioned the full form of ‘UNIGME’. [please see line no. 60]

Comment 11:

The “)” at the last of the first paragraph seems unnecessary, please check and remove it. 

Response 13: Thanks for pointing out the typo. We have removed “)”. [please see line no. 62]

Data and method: 

Comment 12:

In the very first sentence, please write ‘2005-6’ as “2005-06”. Also please provide the survey period of each of the last three rounds of the NFHS.

Response 14: Thank you for your suggestion. We have ensured consistency throughout the manuscript and corrected the year as ‘2005-06’. [please see line nos. 94 and 216]. We have provided survey periods. [please see line nos. 93-94 and response 6]

Comment 13:

Nothing was mentioned about neonatal mortality. Is neonatal mortality, not your dependent variable? You may consider ‘Early initiation’ as well as ‘Pre-lacteal feed’ as predictors, and independent variables as confounders or independent. Moreover, please provide some reference that why you are considering these independent variables in the analysis. 

Response 15: Thank you for your suggestion. We have added a few lines about neonatal mortality in the method section [please see line nos. 137-145] and also included two references by Edmond et al (2008) and Phukan et al (2018). [please see line nos. 137-140]

Results:

Comment 14:

From line number 160-185, nowhere the results (numeric value) are reported. Please report the results in those paragraphs and shorten those. 

Response 16: Thank you for your suggestion. We have included numeric values. [please see line nos. 186-188, 193-195, 200, 202-203, 206-207]

Comment 15:

In Table 3, for the reference categories, you have mentioned ‘@’ as well as ‘R’. Please keep only one.

Response 17: Thank you for your suggestion. We have now used ® to indicate reference category in the Table 3.

Discussion: 

Comment 16:

Too much focus is on the early initiation of breastfeeding and pre-lacteal feeding, and almost nothing is talked about neonatal mortality.

Response 18: Thank you for your suggestion. We have added a paragraph about neonatal mortality. [please see line nos. 365-368, 370-376]

Conclusions:

Comment 17:

The conclusion section is focusing on neonatal mortality only. 

Response 19: Thank you for your suggestion. We have now modified conclusion and included breastfeeding practices as well as pre-lacteal feed. [please see line nos. 386-393, 398-400].

---

## [Editor Report · Decision Letter 1]

11 Sep 2023

PONE-D-22-34236R1First 72-hours after birth: Newborn feeding practices and neonatal mortality in IndiaPLOS ONE

Dear Dr. Ram,

Thank you for submitting your manuscript to PLOS ONE. After careful consideration, we feel that it has merit but does not fully meet PLOS ONE’s publication criteria as it currently stands. Therefore, we invite you to submit a revised version of the manuscript that addresses the points raised during the review process.

We look forward to receiving your revised manuscript.

Kind regards,

Chandan Kumar, Ph.D.

Academic Editor

PLOS ONE

Journal Requirements:

**Additional Editor Comments: **

Thank you for considering the reviewers' comments and suggestions and submitting your revised manuscript to PLOS ONE. However, after careful consideration, we found that the manuscript still needs to address a few minor concerns in order to fully meet PLOS ONE’s publication criteria. The comments are provided in the attached edited copy of the submitted manuscript along with suggested editing in the text using track-change mode. We, therefore, invite you to submit a revised version of the manuscript that addresses the points raised in the attached copy of the manuscript.

---

## [Author Response · Author response to Decision Letter 1]

16 Sep 2023

Thank you to the academic editor for your valuable comments and suggestions. We have responded to each query and have modify the manuscript accordingly.

Note: The line numbers mentioned against every response is based on the clean version of the revised manuscript.

Editor comment 1: It would be better if the authors briefly provide information on the study sample added in the analyses from all the rounds of the NFHS, before discussing the statistical analysis. This becomes more important to add, as there is no section in this paper discussing the sample characteristics, usually mentioned in the Results section. 

Response 1: Thank you for the suggestion. We have included a brief on the same in the method section under a sub head ‘Study sample’ in the revised manuscript [Please see line nos. 106-115].

Editor comment 2: Before it please add a few sentences to describe the nature of the outcome variable, for which the Tobit Regression model has been opted as an appropriate method of analysis. As both the outcome variables seem to be typical dichotomous variables, not censored continuous variables. If it is not so, please briefly describe the rationale for selecting the Tobit regression model. 

Response 2: Thank you for your suggestion. We have explained the same in the revised manuscript. We have now used logistic regression analysis to identify predictors of the early breastfeeding initiation and pre-lacteal feed and modified the text accordingly. [Please see line nos. 147-148 and Table 3]

Editor comment 3: Comparison of or change in the prevalence must be discussed in the same unit for a better understanding of the trend. The authors can show the change while comparing the district prevalence in 2015-16 with the district prevalence in 2019-21. Comparing state prevalence with district prevalence is not useful, although they can be mentioned in the text. 

Please refer to this comment for the interpretation in the paragraph below too.

Response 3: Thank you for the suggestion. We have now modified figure 2 keeping NFHS4 and 5 results. [Please see figure 2]

Editor comment 4: Results must be interpreted in terms of the unit of the study sample. In this study, last-born children/newborns are the study sample unit, so these proportions should refer to the children, not their mothers, despite their numbers being the same. 

Response 4: Thank you for pointing out. We have written the proportion referring the children/newborn babies throughout the text at all appropriate places in the revised manuscript [Please also see line nos. 197, 198, 213, 246, 250, 252-253]. 

Editor comment 5: Please refer to the earlier comments. Results must be written considering the study unit - proportion here refers to children, not the mothers. This must be followed throughout the manuscript.

Response 5: Thanks. The same has been addressed. [Please also see response 4].

Editor comment 6: Received more or less?

Response 6: Thank you for pointing out the missing word error. We have written “received breastmilk early” in the revised manuscript. [Please see line no 208]

Editor comment 7: This doesn't seem to be a correct interpretation of the regression coefficient; these are not the odds ratios.

Please refer to this comment for the following texts wherever it is applicable in the manuscript.

Response 7: Thank you for pointing out the mistake. We have corrected the interpretation of regression coefficient in the revised manuscript at all places. [Please see line nos. 291-304, 306-317]

Editor comment 8: The study sample, i.e., children, constitutes all those born during the five years preceding the respective NFHS. They are not those who were born during the survey period.

Response 8: Thank you for pointing out. We have modified the line. [Please see line nos. 285-287, 315-317]

Editor comment 9: Additionally, authors may also try showing the association between NMR and the coverage of early breastfeeding/pre-lacteal feeding using scatterplots and respective linear regression equations. State-level NMR on y-axis and state-level percentage of early breastfeeding/pre-lacteal feeding on x-axis can be illustrated using scatterplots. 

Response 9: Thank you for your suggestion. We have now added Figure 3 that gives scatterplots showing the association between NMR and the coverage of early breastfeeding/pre-lacteal feeding. State-level NMR are on y-axis and state-level percentage of early breastfeeding/pre-lacteal feeding on x-axis. [Please see Figure 3]

Editor comment 10: This would be good to provide 95% CIs for the NMRs shown in Table 4.

Response 10: Thank you for your suggestion. We have provided 95% CIs for the NMRs shown in Table 4. [Please see Table 4]

Editor comment 11: Authors may thank two anonymous reviewers for their suggestions.

Response 11: Thank you for your suggestion. We have acknowledged two anonymous reviewers and academic editor for their suggestions. [Please see line nos. 449-450]

---

## [Editor Report · Decision Letter 2]

20 Sep 2023

First 72-hours after birth: Newborn feeding practices and neonatal mortality in India

PONE-D-22-34236R2

Dear Dr. Ram,

We’re pleased to inform you that your manuscript has been judged scientifically suitable for publication and will be formally accepted for publication once it meets all outstanding technical requirements.

Kind regards,

Chandan Kumar, Ph.D.

Academic Editor

PLOS ONE

---

## [Editor Report · Acceptance letter]

25 Sep 2023

PONE-D-22-34236R2 

First 72-hours after birth: Newborn feeding practices and neonatal mortality in India 

Dear Dr. Ram:

I'm pleased to inform you that your manuscript has been deemed suitable for publication in PLOS ONE. Congratulations! Your manuscript is now with our production department. 

Kind regards, 

on behalf of

Dr. Chandan Kumar 

Academic Editor

PLOS ONE